# Finding common ground: Understanding and engaging with science mistrust in the Great barrier reef region

Matthew I. Curnock[1]*, Danielle Nembhard[1,2], Rachael Smith[3], Katie Sambrook[4], Elizabeth V. Hobman[5], Aditi Mankad[5], Petina L. Pert[1], Emilee Chamberland[1,6]

1 CSIRO Environment, Australian Tropical Science and Innovation Precinct, James Cook University, Townsville, Queensland, Australia, 2 The Cairns Institute, James Cook University, Smithfield, Cairns, Queensland, Australia, 3 Queensland Department of Environment and Science, Brisbane, Queensland, Australia, 4 C2O Consulting, James Cook University, Townsville, Queensland, Australia, 5 CSIRO Environment, Dutton Park, Brisbane, Queensland, Australia, 6 James Cook University, Townsville, Queensland, Australia

* matt.curnock@csiro.au

**Data Availability Statement:** All data files are available from the CSIRO Data Access Portal database (DOI: https://doi.org/10.25919/6wp0-7m86).

## Abstract

At a time when ambitious environmental management initiatives are required to protect and restore aquatic ecosystems, public trust in the science that underpins environmental policy and decision-making is waning. This decline in public trust coincides with a rise in misinformation, and threatens to undermine public support for, and participation in, environmental protection. Our study investigates the prevalence and predictors of mistrust in science associated with the protection and management of the Great Barrier Reef (GBR) and its catchments. Using survey data from 1,877 residents of the GBR region, we identify environmental values, perceptions, and attitudes that are associated with science mistrust. Our results include a typology of GBR science trust and scepticism. Science-sceptical respondents, representing 31% of our sample, were likely to perceive waterway management decisions as being unfair, felt less responsible, and were less motivated to contribute to improving waterway health than those with greater trust in science. Science-sceptical respondents also had differing perceptions of some threats to waterways, in particular climate change. However, similarities and 'common ground' between respondents with varying levels of trust in science included a shared recognition of the importance of waterways' ecosystem services, and a shared perception of the relative health and problems within their regions' waterways. Our findings can help to break down assumptions about science-sceptical groups in the GBR region and elsewhere. We offer recommendations to guide more constructive engagement that seeks to restore trust and build consensus on mutual goals and pathways to protect vital ecosystem functions and services.

## Introduction

### The role of public trust in environmental science and policy

Public trust in government, public institutions, and science is a fundamental tenet of modern democratic societies. Science underpins technological advancement and humanity's ability to

**Funding:** Funding for this study was provided by the partnership between the Australian Government's Reef Trust and the Great Barrier Reef Foundation, delivered in partnership with CSIRO, the Great Barrier Reef Marine Park Authority, and the Queensland Government's Reef Water Quality Program. The funders had no role in study design, data collection and analysis, decision to publish, or preparation of the manuscript.

**Competing interests:** The authors have declared that no competing interests exist.

understand and overcome complex problems, and trust is placed in scientists to curate complex information and knowledge [1]. Effective environmental policy depends simultaneously on sound science–to incorporate an understanding of system processes and risks, and public participation–to weigh the plurality of human values and trade-offs associated with rules and decisions [2]. Tension at this interface is perpetual; however, in recent decades the politicisation of science in many countries has been accompanied by the polarisation of public views on certain issues and a rise in science mistrust [3–5]. Among environmental management issues, the most prominent example of this phenomenon is anthropogenic climate change, for which the scientific evidence is now irrefutable, yet political and public discourse remains intractably divided [6–8].

Just as scepticism plays a critical role in science, an informed, questioning public is a vital feature of a deliberative and democratic governance system [9]. Where science interacts with government policy, trust in the institutions, actors, and processes is a prerequisite for stakeholder and public engagement and acceptance [10]. However, excessive trust can lead to complacency and overlooked risks, undermining progress towards well-meaning objectives [11,12]. Trust between parties (e.g., policy makers, scientists, stakeholders, and the public) fluctuates over time in response to emerging information and relationship dynamics [13]. Fostering and maintaining 'optimal' levels of trust between institutions and citizens by monitoring and remediating deviations towards too little or too much trust is a desirable strategic objective, as effective resource governance requires parties' adherence to long-term commitments and expectations [11,12].

Optimal trust dynamics between resource managers, stakeholders, rightsholders, scientists and communities do not exist independently of the institutional structures, policy settings, power dynamics, and other contextual factors in a natural resource governance system [12,14,15]. While trust is vital for resilient partnerships and cooperation, there are numerous antecedent and mutually dependent factors that influence governance outcomes. Among these, the integrity of scientific and government institutions and the way they and their agents interact in policy development is crucial [12,15,16], as is public engagement in resource governance via deliberative, transparent, and equitable processes [14,17,18]. In the current period of rapid social and environmental change, effective governance of natural resources requires leaders, resource managers and scientists to invest increasing effort into inclusive and participatory processes for assessing risks and guiding decisions [14,19–21]. Desirable outcomes and objectives of such processes, beyond prudent decision-making and building trust, include social learning and the empowerment of participants, enhanced adaptive capacity and community resilience. Indeed, such processes are worthwhile even if they do not result in measurable improvements in public trust, which is often misplaced as an end-goal, rather than an enabling condition to improved resource and risk governance [14].

## Conceptualising trust for natural resource management

While varying definitions exist in the literature, there is broad agreement that trust is a psychological state of willingness to accept risk or vulnerability, based on one's appraisal and expectations of another party's intentions and behaviour [22–24]. Trust can be placed in people, as well as in non-person entities. One can trust an institution, an object, or a process, though these entities are still implicitly tied to people by virtue of being produced, practiced, or proven by people [24]. Trust can be based on different attributes, including the perception of integrity, competence, benevolence, and charisma. Beyond the perceived characteristics of the trustee, it is also important to acknowledge that the trustor brings with them their own predisposition to be trusting in any given situation (ibid.). People can form a range of trusted relationships at

different scales, representing the diversity of individuals, organisations, and institutions they encounter and/or interact with, each with potentially different qualities, needs and interdependencies [25].

Where trusted relationships coalesce, such as in multi-party resource management or policymaking settings, complex networks of trust can form, creating challenges and opportunities for effective decision-making [12,25]. In this context, four broad types of trust described by Stern and Coleman [24] provide a useful conceptual framework to understand how diverse trust manifests and underpins relationships critical to collaborative natural resource management: (i) *dispositional trust*, which refers to an individual's dispositional and normative tendency to trust (or distrust) institutions, objects or formal roles that have authority or legitimacy, (ii) *rational trust*, which arises from an individual's calculated assessment of an expected benefit or outcome from an exchange or interaction with the trustee, and which requires knowledge of the trustee's ability and/or integrity, (iii) *affinitive trust*, which arises from a perception of the trustee's benevolence, shared values, integrity and/or other personal qualities and character traits, and (iv) *procedural (or systems-based) trust*, which is based on an individual's assessment of control systems, procedures, or rules (rather than an individual or organisation) and which requires knowledge of the legitimacy, fairness, and transparency of the procedure(s) or system. It is proposed that these four types of trust operate across different scales and interact to contribute to a trustor's overall psychological state of trust in real-world situations [24,25].

Different manifestations of the above trust types can affect an individual's motivations and behaviour in myriad ways [24]. For example, a person may have strong dispositional trust for an organisation but lack affinitive trust for its employee(s) and consequently may be motivated to participate in a particular process with those employee(s) to ensure their interests are not misrepresented. Alternately, an individual who has complete trust in an organisation, its processes, and employees might be less motivated to participate in a collaborative process, due to confidence that their interests will be duly considered (ibid.). There are conceptual distinctions too between a *lack of trust*, *mistrust*, and *distrust*. A *lack of trust* can exist in situations where the trustor is unable to make a judgement about the trustworthiness of an individual or other entity, and this may be expressed through apathy or a hesitancy to engage in participatory opportunities. *Mistrust* can be distinguished as a cautious, doubtful, or sceptical attitude towards others. And *distrust* arises from an explicit negative appraisal, and may be expressed through disengagement, the rejection of information, or active opposition [24–27].

## Declining trust in science and the rise of misinformation

A decline in public trust in governments, public institutions, and in science has been documented in many countries over the last few decades [3,28,29]. Due to its inherent complexity and specialised nature, scientific information typically requires synthesis and translation by intermediaries, such as science communicators and journalists, to make it accessible for target audiences and the wider public. The involvement of such intermediaries, however, increases the risk that the scientific information is misinterpreted or framed to serve a particular narrative [29,30]. Once in the public domain, scientific information can be reinterpreted, reframed, and retransmitted by further intermediaries across any number of channels and platforms [30]. Growth in the use of social media as a news source and the increasing fragmentation of news media outlets means that news stories with a scientific component are likely to be viewed by an increasingly smaller audience [31,32]. Despite such risks and challenges, the public communication of science remains essential for its credibility, impact, and the accountability of taxpayer-funded research [30,33].

Misinformation is a growing global problem with often severe consequences. Despite burgeoning research into the phenomenon and its effects, an academic consensus on a precise definition remains elusive due to its many forms and manifestations, including fake news, disinformation, conspiracy theories, and propaganda [34–37]. In the context of science communication, Southwell et al. [36] define *scientific misinformation* as "publicly available information that is misleading or deceptive relative to the best available scientific evidence and that runs contrary to statements by actors or institutions who adhere to scientific principles" (p.98). Misinformation undermines public support for evidence-based policy and has stymied collective action to address problems in multiple domains including the environment, public health, politics, and social inequity [38–40]. The rise of misinformation, accompanying the emergence and growth of social media, is widely attributed as one of the major drivers of declining public trust in science [34,41,42]. However, the causal pathways to becoming mistrustful of science are complex and the relationship between science mistrust and receptivity to misinformation may be one of mutual reinforcement, in which many people turn to misinformation on social media because they are distrustful of the science [34,43].

Scientists and science communicators are also susceptible to pitfalls of misinformation. In a fragmented media landscape scientific news competes for public attention, and science communicators are incentivised to adopt the same tactics that make misinformation appealing, including the use of hyped headlines and overstated results and implications [38]. Other pitfalls faced by scientists include biases favouring the citation and publication of significant results over non-significant results, the misuse of statistics and over-emphasis of $p$-values, and biases in the visualisation and interpretation of results [38,44]. In modern academic culture, the high pressure on scientists to publish frequently and attain notability can erode standards of rigour and ethics in their pursuit of publication metrics and media attention. Meanwhile, a boom in open-access predatory publishers, whose business model relies on collecting authors' fees rather than upholding scientific standards, tempts authors with a shortcut to publication [38,45]. The combined effect of these factors is that scientists can unwittingly contribute to the proliferation of scientific misinformation that undermines public trust in science [38].

## Measuring trust and its associated factors

While in-depth empirical research into the trust process can be performed in the context of interactions between science and everyday citizens, the more commonly reported research is that derived from large-scale population surveys that measure public opinion [29,46]. This type of research often provides headline statistics on comparative levels of public trust in broad sectors and institutions (e.g., scientists, business leaders, politicians, health authorities), and enables comparisons of different cohorts (e.g., by demography, political orientation) and countries, and the assessment of temporal changes, including in relation to significant events (e.g., the COVID-19 pandemic) [46–48]. While trust is ideally measured across multiple dimensions, limitations inherent to survey design result in many studies employing only a single item to measure trust within a specific context, and the variation across such studies limits their comparability [29]. Single-item, generic measures of public trust in science are nonetheless useful for understanding associated factors and for monitoring trends in an applied context [24,29,46].

In the context of environmental management, public and stakeholder trust is a focal topic for many case studies that explore the relationships between trust and pro-environmental behaviours, risk perceptions, and support for management initiatives and policy [24,49]. Empirical studies across a variety of contexts have shown that trust in science and management institutions has a strong influence on perceptions of environmental threats and risks,

which in turn affects support for protective measures and the adoption of conservation-related behaviours [e.g., 50–54]. The relationship between trust and personal environmental values, however, is more complex. While an affinity for nature has been shown to correlate positively with support for environmental protection and pro-environmental behaviour [55,56], personal environmental values appear to have an indirect relationship with trust, serving as an antecedent factor in the trust-forming process [57].

## Research to inform trust-building engagement

Increased recognition of the importance of trust in environmental management and of declining public trust in science has prompted growth in research to understand underlying factors and provide advice to scientists, communicators and resource managers who seek to build and maintain trust for improved management outcomes. Much of this research focuses on public and stakeholder engagement [e.g., 58–63], and while engagement on its own is insufficient to build and maintain trust due to its multi-dimensional nature and emergence in a societal context, it nonetheless plays a critical role in fostering trust within a wider governance system [12,14,18]. Some communication and engagement approaches that can contribute to building trust in a wide range of contexts and domains include:

- Engagement by relatable and credible leaders and spokespeople who are regarded as 'trusted messengers' [60,64–68, among others].

- Demonstration of empathy, commonality of values and social identity, and a shared vision for the future [50,59,60,62,66,69, among others].

- Clear and relatable message framing (using appropriate language, metaphors, and images) of relevant issues, risks, and opportunities, to build a shared understanding of problems and consensus on objectives or solutions, and pathways to achieving mutual goals [65,68,70,71, among others].

- Demonstration of scientific consensus, scientific expertise, and competency, as well as robustness of scientific methods and data [50,63,72–74, among others].

- Honesty and transparency about knowledge gaps, uncertainties, risks, and expected outcomes [50,59,62,63,65,75, among others].

- Consistent demonstration of scientific integrity, and impartiality regarding its contribution to policy development and decision-making processes [12,14,15,62,76,77, among others].

- Demonstration of a long-term commitment to participatory engagement, two-way information sharing, fairness, and deliberative governance [14,63,78–80, among others].

Effectively addressing points two and three above requires an in-depth understanding of the target audience, which can emerge from a history of direct engagement, shared experiences, and/or peer networks. For cases in which there is an incomplete understanding of a cohort's values, perceptions, and normative beliefs relevant to a specific context or issue, social research utilising surveys can be useful for eliciting insights about 'common ground' on which constructive engagement may be built.

## Case study context–the great barrier reef

The Great Barrier Reef (GBR; the Reef) is an iconic, complex, and dynamic socio-ecological system. Its diverse marine ecosystems are connected to adjacent coastal habitats and river catchments, which provide passageways for mobile and migratory species, and drain terrestrial

runoff into the GBR lagoon, linking the Reef with human activities in its catchment areas [81,82]. However, the GBR is threatened by a range of human-caused pressures that are affecting its ecology, values, and resilience. These pressures include anthropogenic climate change, which contributes to marine heatwaves and the intensification of extreme weather events [83], large-scale outbreaks of coral-eating crown-of-thorns starfish, poor coastal and inshore water quality from sediments, pesticides and nutrients in land-based runoff, and direct human uses such as illegal fishing [81,82,84]. Coastal and catchment habitats in the GBR region too, are under pressure from increasingly frequent and severe climatic events (e.g., flooding, heatwaves, bushfires), as well as degradation from land developments and uses that exacerbate vegetation loss and soil erosion [81,85,86].

The high value that humans place on the GBR is evident in the efforts that have been made to protect and conserve its ecological integrity and heritage, nationally through the *Great Barrier Reef Marine Park Act 1975*, globally through its 1981 World Heritage listing, and in response to more recent events and pressures through the *Reef 2050 Long-Term Sustainability Plan* [87]. Recognising that climate change is a global issue that cannot be managed at a local or regional scale, the State and Federal government authorities responsible for protection and management of the GBR have largely focused on mitigating local and regional pressures, including improving catchment water quality through policies aimed to improve agricultural practices, direct interventions to limit impacts from crown-of-thorns starfish outbreaks, and regulation and enforcement of commercial and recreational fishing [82,87,88]. Scientific evidence has played a critical role in guiding the governments' policy development, decision-making and investments, through demonstrating the condition and trend of ecosystems, the extent of current and projected impacts, the source of stressors, and importantly, the actions that are required to mitigate pressures and protect ecological values and processes [e.g., 81,89–91].

Like other major policy initiatives, water quality improvement from the GBR's catchments requires community support and stakeholder participation to be effective. Management efforts to date under the *Reef 2050 Water Quality Improvement Plan* [88] and other policy instruments have had a major focus on landholder and industry initiatives to reduce sediment, nutrients, and pesticide runoff, with incentives to promote voluntary changes in agricultural land use practices and to restore degraded ecosystems. However, the uptake of land use practice change has been slow, and policy initiatives have faced strong and sustained opposition from some groups [92–95].

Within the last decade, the science underpinning Reef and water quality management has been the subject of increased scrutiny and criticism, as well as misinformation on social media and partisan news media platforms [94,96–98]. A series of high-profile events within the GBR have attracted extensive media coverage, generating international interest and arousing public sentiment about the Reef's health and protection. Such events have included government approvals for major port developments in 2014 [99], mass coral bleaching events of unprecedented scale and severity in 2016 and 2017 [100–102], and reactive monitoring missions by the UNESCO World Heritage Centre in 2012 and 2022 to assess the state of conservation of the GBR World Heritage property and consider its inclusion on the list of World Heritage properties "in Danger" [96,97]. These events, particularly the mass coral bleaching, have been accompanied by both sensationalised media representations of the GBR's demise [e.g., 103], and misinformation that claims the GBR is in good health and is unthreatened by climate change and poor water quality [97,98,104].

Among the narratives promulgated by some media platforms, challenges to the veracity of GBR science and the integrity of its scientific institutions have become a frequent feature, garnering sufficient public and political interest to prompt an Australian Senate Committee Inquiry in 2020 into the evidence base underpinning regulation of farm practices that impact

GBR water quality. While the adverse claims about this evidence base and the quality of GBR science were ultimately dismissed by the Committee [105], misinformation that sows doubt about the quality and integrity of Reef and water quality science continues to proliferate in the region and nationally, particularly as the health of the GBR epitomises characteristics of the broader climate change debate [98].

While the direct effects of misinformation on public trust in GBR science and management are not well understood, there are concerns that declining public trust in science could undermine political support for efforts to protect and restore its waterways at a critical time. Indications from long-term social surveys in the region suggest that while Reef management and scientific institutions are the most trusted sources of information about the GBR, residents' trust in these institutions has decreased since 2017 [106,107].

## Research aims and questions

The rationale for our case study was to better understand and articulate the characteristics of science mistrust in the GBR region. Using statistical analyses of data from a survey of residents in the catchments of the GBR region, and by adopting a typological approach, we categorised respondents into groups based on their levels of stated trust in the science about waterway health and management (*trust in science*) and compared their responses to a broad set of rating-scale questions about waterway values, behaviours, perceptions, and attitudes. From the analysis we sought to identify commonalities, or 'common ground' that can be leveraged for more effective engagement with those stakeholders and communities who are mistrustful (or distrustful) of the science underpinning GBR and water quality management. Specific research questions included:

i. What is the prevalence of mistrust in science associated with waterway health and management among residents of the GBR catchment area?

ii. What environmental values, activities, perceptions, and attitudes are associated with (or are potential predictors) of trust and mistrust in such science?

iii. What values, perceptions and attitudes do 'science trusting' and 'science sceptical' groups of residents have in common?

iv. How might such commonalities serve as a basis for trust-building communication and engagement?

Findings from this case study are intended to be applied in the GBR region and potentially elsewhere by scientists, science communicators, leaders, and resource managers who seek to improve their engagement with science-sceptical groups to advance environmental protection goals. While insights about distinctive characteristics of science-sceptical and science-trusting groups can be informative, an improved understanding of the 'common ground' between parties is more likely to serve as a basis for productive dialogues.

## Materials and methods

### Survey dataset and data collection

We used a social survey dataset of 1,877 residents of the GBR catchment region. The survey was conducted in November 2021 by four Regional Report Card (RRC) partnerships, each representing a major catchment of the GBR, in collaboration with the research team and management agency staff from the Queensland Government. Four different survey instruments were deployed simultaneously by the RRC partnerships, tailored to each region's characteristics;

however, many of the questions were identical across the four surveys, enabling cross-regional comparisons. The purpose of the survey was to provide a baseline for long-term monitoring of 'human dimension' indicators that would help to evaluate government management agencies' progress towards achieving a set of objectives in the Reef 2050 Long Term Sustainability Plan [87], and to inform adaptive management of waterways, including strategic communication with communities and stakeholder engagement in the regional catchments. The surveys were co-designed via a participatory process involving officers from each of the Regional Report Card partnerships, relevant management agency staff, and social scientists. Details of the co-design process, the objectives and constructs underpinning the survey metrics, data collection, and the curation of the survey dataset are documented in a technical report [108]. The dataset itself is publicly accessible via the CSIRO Data Access Portal ('*Great Barrier Reef Catchment Regional Waterway Partnerships Baseline Social Surveys*'; CC-BY-NC 4.0 Licence; DOI: 10.25919/52yr-rg31 [109]).

Respondents participated in the survey online and were recruited via one of two possible pathways: either (a) as part of an online panel administered by a market research provider, or (b) via regionally targeted advertising through local print media (with QR codes) and social media (via Facebook[TM]). The panel consisted of volunteers who periodically undertake market research surveys to earn credits (e.g., redeemable loyalty card points), and are selected to participate in surveys based on their demography and location. The panel provider used for this survey was an accredited member of the Australian Data and Insights Association, holding ISO 20252 certification for 'Market, opinion, and social research'. Such panel providers typically draw on suitable respondents from a large pool of members in metropolitan areas; however, in regional areas the pool can be smaller. In the four GBR catchment regions, the number of eligible respondents was estimated (by the provider) to range between 100 and 400 people per region. To achieve a desired sample from each region that would enable robust statistical comparisons, the survey recruitment was supplemented using paid social media advertisements (targeted within regional postcodes), as well as via local newspapers and other regional channels. Each RRC partnership was responsible for the supplementary recruitment in their region. Two of the regions used a prize draw (local tour voucher) to attract a larger pool of respondents. An underlying principle of the supplementary recruitment was to avoid sampling bias, thus paid social media advertisements were used instead of 'organic' sharing of the survey link, to avoid over-representation of sympathetic respondents or 'friends' of the RRC partnerships, and to capture a representative and diverse field of community views on the survey topics. The abovementioned technical report [108] provides further details of the survey methods, data returns, geographic boundaries, and respondent demography in each region.

## Survey ethics

Participation in the online survey was voluntary and respondents remained anonymous. An introductory information page outlined the purpose of the survey, the lead organisations and funding source, the intended uses of the survey data, potential risks and benefits associated with participation, confidentiality and privacy provisions, as well as details of the ethical clearance and contacts. Informed consent by respondents was indicated via a tick box ("Do you consent to take part in the survey?"; "Yes" or "No") at the end of the introductory information, prior to commencement of the survey questions. The study, its procedures, the survey questions, the introductory information, and the means of obtaining prior consent were reviewed and approved by CSIRO's Social Science and Human Research Ethics Committee (CSSHREC; Approval number 140/21), in accordance with Australia's *National Statement on Ethical Conduct in Human Research* (2007), prior to survey commencement.

## Survey questions

The survey's main focal topic was *regional waterways*, which were defined in the preamble as encompassing *freshwater systems* ('all creeks, streams, rivers, lakes, dams, and wetlands'), *estuaries* ('the lower reaches of creeks and rivers that are tidal where salt and freshwater mix'), and *marine habitats* ('coastal waters including beaches and islands extending to the Great Barrier Reef') within the RRC boundary. A figure depicting these different zones within each region's boundaries was also provided as a visual reference [108].

The surveys focussed on residents' (i) *uses*, *benefits and values* associated with waterways, (ii) their *perceptions* of the waterways' health, problems, and threats, (iii) participation in waterway *stewardship* and enabling factors such as their motivation and capacity, and (iv) their perceptions of waterway *governance*, including support and trust for management institutions, and their trust in the science underpinning waterway management (i.e., "*I trust the science about waterway health and management*"; hereafter referred to as '*trust in science*'). Most of the survey questions, including that for *trust in science* above, utilised ten-point Likert-type scales, representing their relative agreement with a statement (1 = 'very strongly disagree'; 10 = 'very strongly agree'), or the extent to which they value a characteristic of their region's waterways (1 = 'I don't value this at all'; 10 = 'I value this extremely highly'). Some questions used shorter response scale options with defined response categories. For example, perceptions of the health of different ecological habitats were elicited via a three-point scale (1 = 'in poor health', 2 = 'in fair health', 3 = 'in good health'), and a five-point scale was used for eliciting perceptions of the problems and threats affecting different waterway habitats (1 = 'does not represent a problem/threat at all', 2 = 'a small problem/minor threat', 3 = 'a moderate problem/threat', 4 = 'a big problem/serious threat', 5 = 'represents a very big problem/extremely serious threat'). An 'I don't know' option was provided for questions of this type where relevant. Other questions in the survey utilised 'tick box' categories (e.g. age, gender, employment sector, participation in specific waterway recreation and stewardship activities).

## Basis of typology

Our typology in the following results was based on respondents' numeric ratings indicating their level of agreement or disagreement with the above *trust in science* statement. We assigned respondents a category based on their numeric responses. Respondents who gave ratings of one or two (indicating strong disagreement with the statement) were assigned to the 'Strongly Sceptical' group; respondents who gave ratings of three to five were classified as 'Mildly Sceptical'; those who gave ratings of six to eight were classified as 'Mildly Trusting'; and those who gave ratings of nine or ten (indicating strong agreement) were assigned to the 'Strongly Trusting' group.

Typological comparisons of the four groups were made using a two-step process. First, we used regression tests to determine any statistically significant relationships for a range of potential predictors with *trust in science*. We then examined any statistically significant results by comparing the mean ratings (±SE), to make a qualitative assessment of the differences for descriptive purposes. In some cases, a meaningful difference between the typology groups' mean scores was not apparent even when a statistically significant regression was found.

## Statistical tests

Statistical tests were performed using R Statistical Software (v4.3.1) [110]. To identify variables associated with science mistrust we performed a series of ordinal logistic regression analyses with respondents' rated level of agreement (1–10, as per above) with the *trust in science* statement as the response variable. Ordinal logistic regression analyses are useful for determining the likelihood that an ordered, categorical response variable can be explained by variation in

the predictor variables [111–113]. The Cumulative Logit Model (CLM), also known as Proportional Odds Model (POM), assumes that the effect of the predictor variables on the response variable is the same for all categories of the response variable [114,115]. The CLM is the most widely used model and enables the evaluation of the correlation between the response and predictor variables without having to fit separate models for each category of the response variable [112]. Separate regression models were run for each survey question that was treated as a potential predictor.

We tested for the assumptions of ordinal logistic regressions: (i) multicollinearity, (ii) proportional odds, and (iii) goodness of fit. The validity of ordinal regression models relies on the absence of high multicollinearity. To check for multicollinearity, we calculated variance inflation factors (VIFs), where higher VIF values indicate higher multicollinearity. In general, VIF values above five are of concern while a VIF greater than 10 indicates severe multicollinearity. Therefore, only variables with VIF values less than five were included in the final ordinal regression analyses for each survey question. The proportional odds assumption, also known as the assumption of parallel lines, states that the odds of one unit change in the predictor variable influencing the response variable are constant across all levels of the dependent variable [116]. The Brant Test was applied to test the assumption of parallel lines [117]. It should be noted that there are instances when these statistical tests falsely reject the null hypothesis that the assumption is satisfied, leading to incorrect conclusions that the analyses are invalid [118]. As such, we used exploratory graphical analyses of residuals (residual plots) along with statistical hypothesis tests to evaluate the goodness-of-fit of the assumptions' models [119]. We also used Pearson tests to evaluate goodness-of-fit, comparing the observed and predicted frequencies of the outcome variable to determine the overall fit of the regression models [120].

## Results

### Sample description and prevalence of science mistrust

From the total sample of 1,877 respondents, 31% (n = 579) gave a rating of five or lower on the ten-point scale, indicating disagreement with the statement 'I trust the science about waterway health and management' ('*trust in science*'). Among these, from the total sample, 8% (n = 141) were assigned to the 'Strongly Sceptical' (SS) group, and 23% (n = 438) were classified as 'Mildly Sceptical' (MS). Among those who gave ratings of six or above on the response scale, indicating degrees of *trust in science*, 45% of the total sample (n = 839) were classified as 'Mildly Trusting' (MT) and 24% (n = 459) were assigned to the 'Strongly Trusting' (ST) group.

A comparison of demographic characteristics across the four trust groups (Table 1) revealed a higher proportion of males comprising the SS group (69%), while the MS, MT and ST groups had comparable representation of genders, albeit with a higher proportion of female respondents that was inherent in the overall survey sample (54% were female). While the median age (in categories) was the same across the four groups, the SS group had a higher proportion of respondents in age categories over 55 years (49%), when compared with the MS (42%), MT (41%), and ST (35%) groups. Similarly, SS and MS respondents had lived in their region longer on average (31 years and 26 years respectively) than MT (24 years) and ST respondents (20 years). The SS group was also represented by a higher proportion of respondents who were employed in primary industries (28%), including the agriculture sector (16%), than the other three trust groups (Table 1; see also S1 Table).

### Waterway values

Ordinal regression results showing the relationship between respondent ratings for *trust in science* and a series of value statements relating to aspects of regional waterways (1 = 'I don't

**Table 1. Demographic characteristics of survey respondents in four categories representing their varying '*trust in the science about waterway health and management*' in the GBR region (n = 1877).**

| Trust categories and demographic characteristics | Strongly Sceptical (SS) | Mildly Sceptical (MS) | Mildly Trusting (MT) | Strongly Trusting (ST) |
|---|---|---|---|---|
| Number of respondents (proportion of total sample) | 141 (8%) | 438 (23%) | 839 (45%) | 459 (24%) |
| Female: Male (ratio; %) | 31: 69 | 55: 45 | 56: 44 | 59: 41 |
| Median age in categories | 45–54 years | 45–54 years | 45–54 years | 45–54 years |
| Proportion of respondents 55 years & older | 49% | 42% | 41% | 35% |
| Mean duration lived within the region | 31 years | 26 years | 24 years | 20 years |
| Proportion employed in primary industries (Agriculture) (*Mining*) | 28% (16%) (*11%*) | 16% (4%) (*11%*) | 11% (3%) (*8%*) | 11% (3%) (*7%*) |
| Proportion dependent on regional waterways for some of household income | 30% | 23% | 22% | 31% |

value this at all'; 10 = 'I value this extremely highly') are presented in Fig 1 (panel a), with additional details for each of the test results reported in S2 Table. Significant positive relationships were found for *First Nations heritage* ('the waterways have rich heritage to First Nations people/Traditional Owners'), *local recreation* ('the waterways offer a place for local residents to enjoy recreation activities'), *existence value* ('the fact that the waterways exist, even if I don't use or directly benefit from them'), *tourism attraction* ('the waterways are an important attraction for tourists visiting the region'), and *iconic status* ('our waterways are recognised nationally and internationally for their iconic status, e.g., World Heritage, RAMSAR sites'). These results indicate that respondents who placed a higher *trust in science* were more likely to value these aspects of their region's waterways to a higher degree than respondents who had a lower *trust in science*. A significant negative relationship was found between *trust in science* and *local agriculture* ('the waterways support local agriculture') indicating that respondents who had a lower *trust in science* were more likely to value local agriculture uses of waterways higher than those with greater *trust in science*.

To inform our descriptive typology, it was important to carefully examine the response patterns for the four trust groups. While the abovementioned regressions were statistically significant, such results do not always reflect a linear relationship between response and predictor variables, nor do they indicate low or high mean ratings for any group *per se*. When comparing the mean ratings between the four groups for *First Nations heritage* values of regional waterways, we see that the relationship to *trust in science* does appear to be linear. The SS group's mean rating was near the middle of the response scale (5.56 ± 0.28 SE), while the ST group's mean rating was relatively high (8.14 ± 0.11 SE), and the MS and MT groups' mean scores fell in between (Fig 1B). Comparing other waterway values that were correlated significantly with *trust in science*, we observe that while the ST group gave a very high mean rating for their regional waterways' *existence value* (9.46 ± 0.05 SE), the mean ratings for all groups were relatively high (i.e., all above 8.0), indicating a shared strong value. Similarly, we observe that the mean ratings for all four trust groups' values for waterways supporting *local agriculture* are very similar (all between 6.89 and 7.28 on the ten-point scale; Fig 1B). The mean scores corresponding to regression results for other waterway values (S2 Table) show small and/or non-significant differences, indicating similarity in the importance attributed to these waterway values between the four trust groups.

## Governance perceptions

Fig 1 (panel c) shows ordinal regression results between respondents' *trust in science* and their ratings for a series of statements reflecting their perceptions and attitudes towards waterway

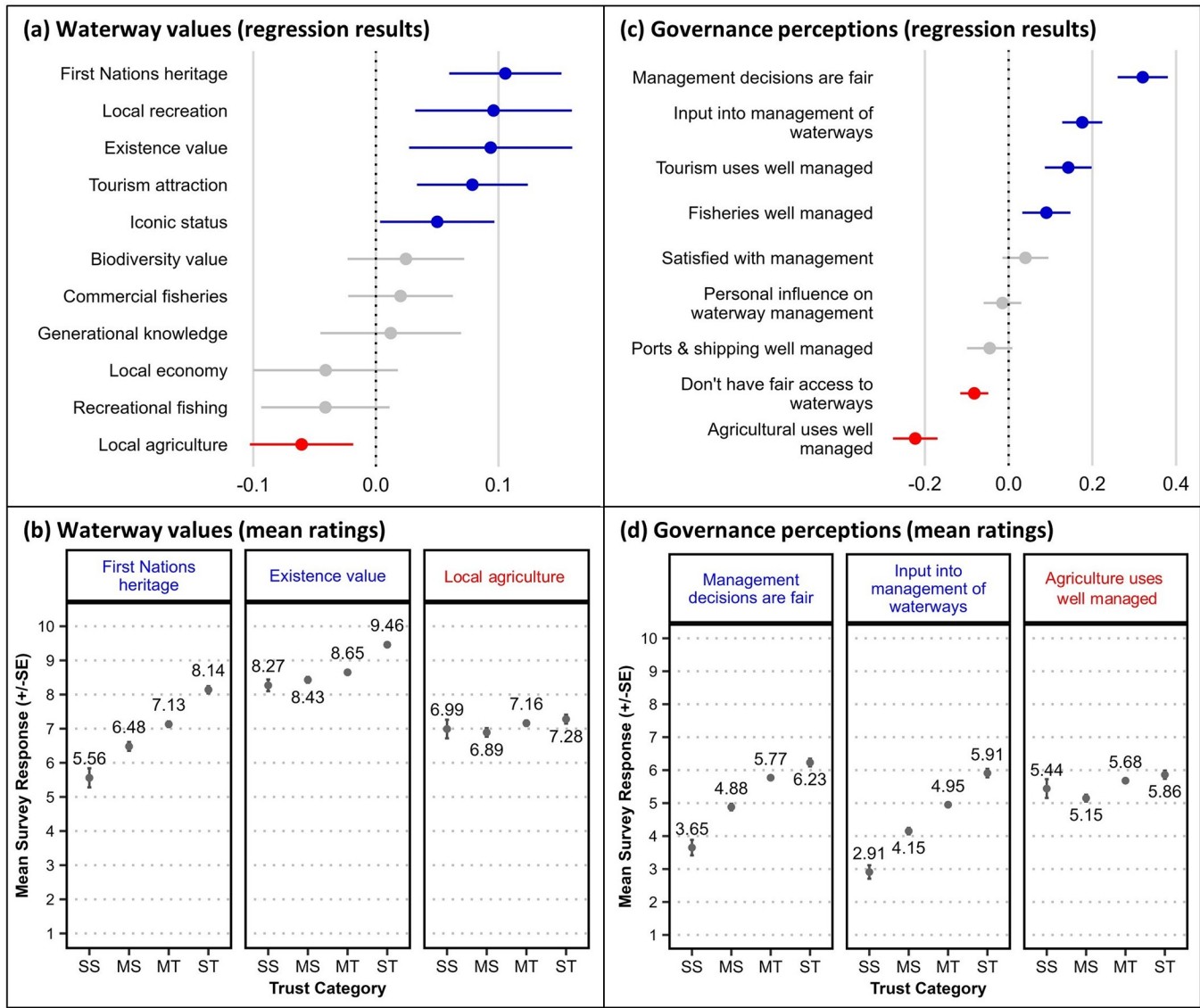

**Fig 1. Relationships between survey respondents' *trust in science* and their values attributed to regional waterways, and perceptions of waterway governance.** Upper panels show ordinal regression test results plotting survey respondents' *trust in science* with (a) their values attributed to regional waterways, and (c) their perceptions of waterway governance (n = 1,877). Regression coefficients [dots] and standard error [SE] bars show statistical significance of the relationship where intersection of the SE bar with zero indicates lack of significance. Blue colouring indicates a significant positive relationship and red colouring indicates a significant negative relationship. Lower panels (b, d) show the mean rating scores (±SE) from four groups with differing stated trust in science (SS = strongly sceptical, MS = mildly sceptical, MT = mildly trusting, ST = strongly trusting) for selected survey items with a significant regression result.

governance (1 = 'very strongly disagree'; 10 = 'very strongly agree'). We found significant positive relationships between respondents' *trust in science* and their perceptions that *management decisions are fair* ('I think that decisions about managing local waterways are made in a fair way'), that they are able to provide *input into management of waterways* ('I feel able to have input into the management of waterways in my region if I choose to'), and that uses of regional waterways by the *tourism* and *fisheries* sectors *are well managed* ('I think that tourism uses of waterways in our region are well managed'; 'I think that the fisheries in our region are well managed'). Significant negative relationships were found between *trust in science* and

respondents' perceptions that they *don't have fair access to waterways* ('I do not have fair access to all the waterways in my region that I would like to use') and that *agriculture uses* (of regional waterways) are *well managed* ('I think that agriculture uses of waterways in our region are well managed').

A comparison of the mean ratings from our four trust groups for the above governance perceptions (Fig 1D and S3 Table) shows a relatively linear, positive relationship between *trust in science* and perceptions that *management decisions are fair* and that respondents can provide *input into management of waterways*. The mean ratings for SS respondents for these variables were particularly low (3.65 and 2.91 out of 10, respectively), indicating that GBR residents with low *trust in science* do not perceive sufficient opportunities to contribute to management decisions affecting their regional waterways, and have a dim view of those management institutions' procedural fairness.

Despite a significant negative correlation between *trust in science* and the perception that *agricultural uses of waterways are well managed*, a comparison of the trust groups' mean ratings shows only minor differences in such perceptions between science sceptical and trusting respondents (Fig 1D). Similarly, while significant positive relationships were found for perceptions of *tourism uses* and *fisheries* (Fig 1C), the spread of mean ratings between the trust groups were smaller than those for other predictor variables, and there was no discernible difference between the SS and MS groups (*tourism uses* SS = 5.50 ± 0.227; MS = 5.47 ± 0.097; MT = 6.33 ± 0.057; ST = 7.00 ± 0.101, and *fisheries* SS = 4.74 ± 0.249; MS = 4.93 ± 0.090; MT = 5.90 ± 0.063; ST = 6.42 ± 0.110; S3 Table).

A significant relationship was not found between *trust in science* and respondents' *satisfaction with waterway management* ('overall, I feel satisfied with how local waterways are managed') despite an incremental rise in the mean satisfaction ratings across the four groups coinciding with increasing *trust in science* (SS = 3.65 ± 0.235; MS = 4.88 ± 0.098; MT = 5.77 ± 0.063; ST = 6.23 ± 0.116). Similarly, the mean ratings for respondents' *personal influence on waterway management* ('I feel I personally have some influence over how local waterways are managed') rose incrementally across the trust groups (SS = 2.60 ± 0.193; MS = 3.73 ± 0.095; MT = 4.29 ± 0.073; ST = 4.81 ± 0.130; S3 Table), noting that the mean ratings for all groups were relatively low.

## Waterway uses and benefits

Participation in a range of waterway recreation activities was elicited ('when visiting all the different waterways in the region in the past 12 months, what recreational activities have you participated in?'), with respondents selecting applicable activities from a list. Logistic regression tests found significant positive relationships between *trust in science* and *appreciating nature* ('wildlife watching and appreciating nature') and participating in in-water activities such as *snorkelling and diving* and *swimming* (Fig 2A). Significant negative relationships were found between *trust in science* and participation in *fishing* and *motorised watersports* (e.g. water skiing and jet skiing). A comparison of the mean ratings for the four trust groups revealed appreciably higher participation in wildlife watching and *appreciating nature* among ST respondents, and in *fishing* among SS respondents when compared to the other three groups, but little apparent difference in participation in *motorised watersports* between the four groups, despite the significant regression result (Fig 2B; see also S4 Table).

The importance of specific personal benefits from regional waterways (i.e., individual benefits derived from ecosystem services) were rated by respondents on a ten-point agreement scale. *Experiencing nature* ('the waterways are important for allowing me to experience, appreciate and interact with the natural environment') and providing a *domestic water supply* ('the

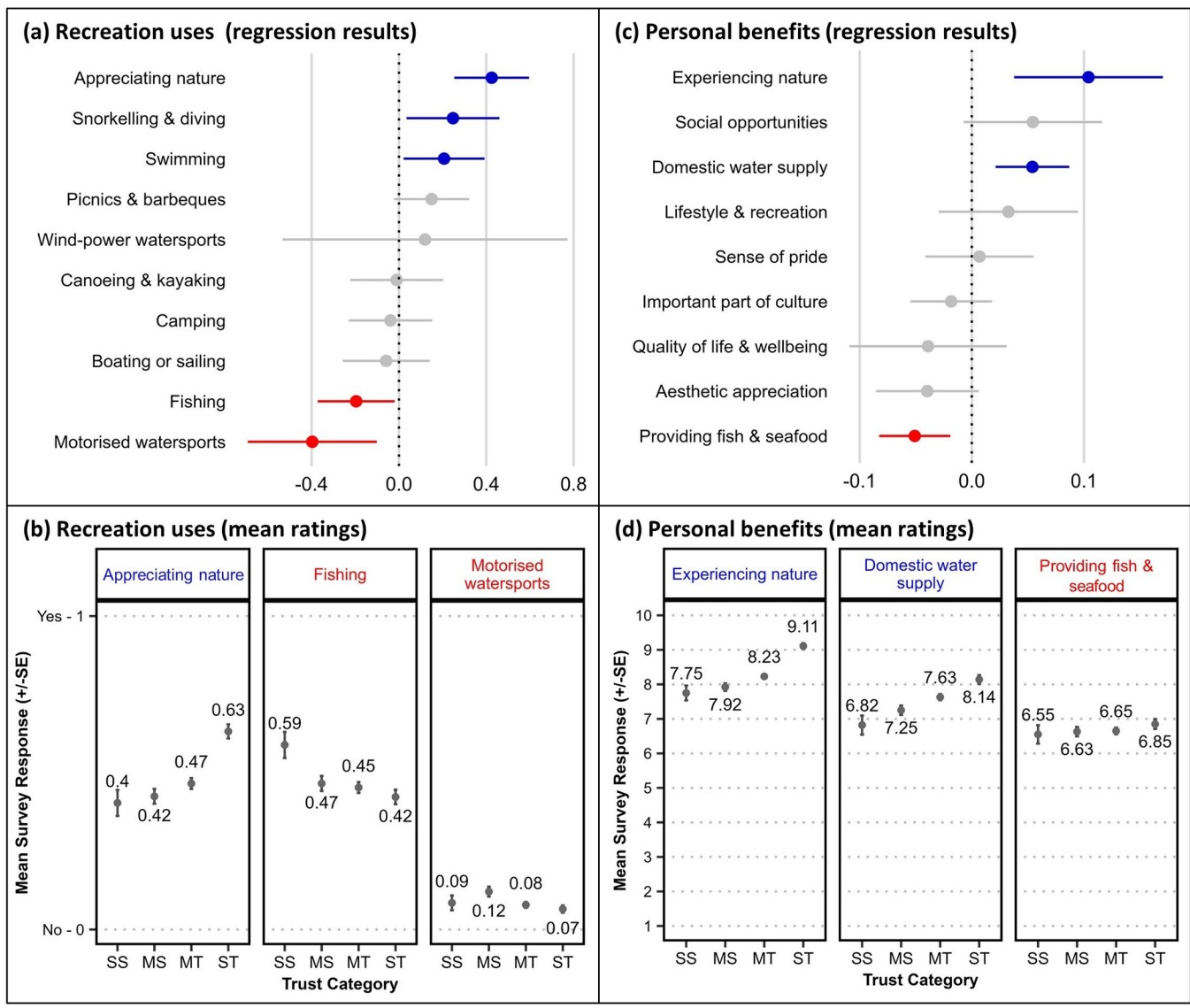

**Fig 2. Relationships between survey respondents' *trust in science* and their recreation uses of waterways, and perceptions of waterway benefits.** Upper panels show ordinal regression test results plotting survey respondents' *trust in science* with (a) their recreation uses of waterways of regional waterways, and (c) their perceptions of personal benefits derived from waterways (n = 1,877). Regression coefficients [dots] and standard error [SE] bars show statistical significance of the relationship where intersection of the SE bar with zero indicates lack of significance. Blue colouring indicates a significant positive relationship and red colouring indicates a significant negative relationship. Lower panels (b, d) show the mean rating scores (±SE) from four groups with differing stated trust in science (SS = strongly sceptical, MS = mildly sceptical, MT = mildly trusting, ST = strongly trusting) for selected survey items with a significant regression result.

waterways are an important source of my water supply for drinking and household use') were found to have a significant positive relationship with *trust in science*, while *providing fish and seafood* ('waterways in the region are important for providing fresh fish and seafood for me to eat') was negatively correlated (Fig 2C). The plotted mean ratings for the four trust groups show perceptible but minor differences for *experiencing nature* and *domestic water supply*, and little discernability for *providing fish and seafood* (Fig 2D; see also S5 Table). The minor differences for these few variables, and the absence of significant relationships for other items indicates a broadly shared appreciation for ecosystem-derived benefits between science sceptical and trusting groups.

## Participation in waterway stewardship

Respondents indicated their participation in a set of activities associated with environmental stewardship in or around their region's waterways ('For the following questions, we would like to ask you about several personal actions that are intended to improve waterway health. Which of the following do you personally do?'). Regression tests of a binary response option (yes or no) found small but statistically significant relationships between *trust in science* and *contributing to environmental monitoring* ('contribute to environmental monitoring programs, e.g., by participating in data collection, or reporting wildlife sightings'; positively correlated) and *reporting suspicious activities* ('report suspicious activity to relevant authorities, e.g., illegal dumping, illegal fishing practices, chemical or oil spills'; negatively correlated; Fig 3A). Small distinctions were apparent between the trust groups' mean scores for these two variables (Fig 3B; see also S6 Table); however, for the purposes of a typological comparison these distinctions did not indicate a substantive difference between science sceptical and science trusting respondents in their participation in environmental stewardship.

## Stewardship enablers and barriers

Personal capacity and motivational factors associated with environmental stewardship were elicited from respondents via their agreement/disagreement ratings (1–10 scale, as above) for a series of statements. Regression tests found significant positive relationships between *trust in science* and *feeling responsible* ('I feel a sense of responsibility to help improve waterway health'), *wanting to do more to help* ('I want to do more to help improve waterway health in my region') and *personal efficacy* ('I can make a personal difference to improving waterway health in my region'; Fig 3C). For these three motivational factors, the distinctions between the four trust groups' mean ratings indicate that science sceptical (both SS and MS) respondents feel less responsible, less motivated, and less empowered to make an effective personal contribution to the health of waterways in their region than science trusting respondents, and ST respondents in particular (Fig 3D).

Ratings for statements reflecting normative beliefs about community participation and support for waterway stewardship, such as *local residents taking action* ('many local residents in my region are taking action to improve waterway health') and *local residents support action* ('local residents in my region are supportive of taking action to improve waterway health') revealed no significant relationship with *trust in science*. Likewise, significant relationships were not found for statements reflecting respondents' capacity in terms of *time* ('I don't have enough time to contribute to improving waterway health in my region') and *knowledge* ('I don't know how I could contribute to improving waterway health in my region') to contribute to waterway stewardship, nor for a statement reflecting their *hope for the future* of their region's waterways ('I feel hopeful about the future health of waterways in my region'; Fig 3C; S7 Table).

## Perceptions of waterway health, problems, and threats

Respondents' perceptions of the relative health of different waterway habitats from the catchment to offshore coral reefs (indicated via a three-point scale as described in methods) were largely unrelated to their *trust in science* (Fig 4A). Two exceptions were a significant (but slight) positive correlation with perceptions of the health of *beaches and the coast*, and a significant negative correlation with that of *inshore coral reefs*. For the latter result a distinct difference in mean ratings was evident, indicating that SS respondents perceived inshore coral reefs to be in much better health than the other trust groups (Fig 4D; S8 Table).

Perceptions of potential problems manifested in regional waterways (elicited via a five-point scaled response as per methods) were also mostly unrelated to *trust in science* (Fig 4B);

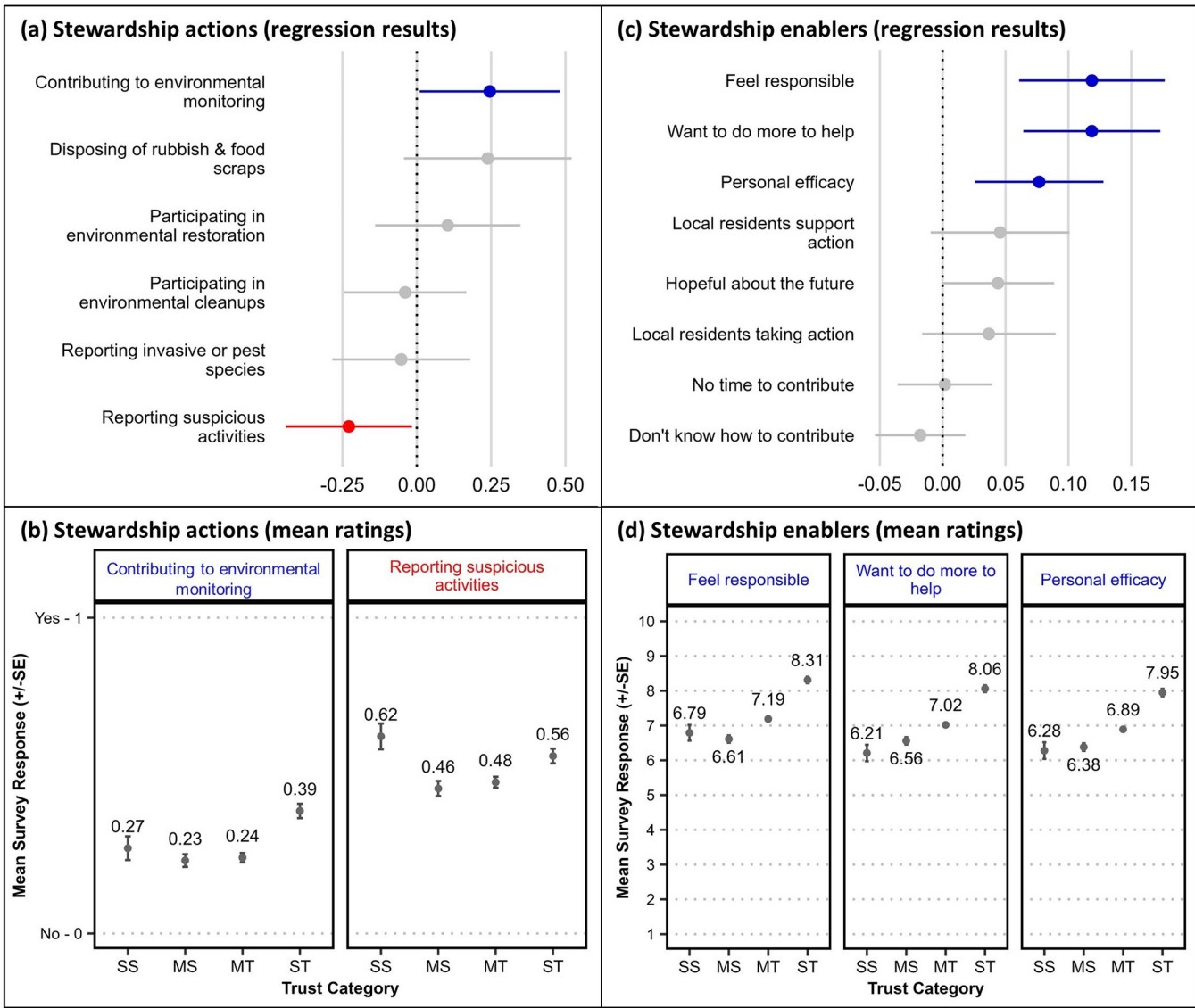

**Fig 3. Relationships between survey respondents' *trust in science* and their self-reported waterway stewardship, and stewardship enabling factors.** Upper panels show ordinal regression test results plotting survey respondents' *trust in science* with (a) their self-reported participation in stewardship actions, and (c) personal capacity and motivational 'stewardship enabling' factors (n = 1,877). Regression coefficients [dots] and standard error [SE] bars show statistical significance of the relationship where intersection of the SE bar with zero indicates lack of significance. Blue colouring indicates a significant positive relationship and red colouring indicates a significant negative relationship. Lower panels (b, d) show the mean rating scores (±SE) from four groups with differing stated trust in science (SS = strongly sceptical, MS = mildly sceptical, MT = mildly trusting, ST = strongly trusting) for selected survey items with a significant regression result.

however, significant positive correlations were found for perceptions that *chemical pollutants* ('chemical pollutants, e.g., pesticides, PFAS') and *riverbank erosion* were problematic issues in regional waterways. When examining the mean ratings (Fig 4D; S9 Table), discernible differences between the trust groups were evident, in which the problem perception for both items was lowest among SS respondents.

A strong and significant positive relationship was found between *trust in science* and the perception of *climate change* as a threat to waterways, with an apparent polarity between the trust groups' mean rating scores (Fig 4C and 4D). For nearly all other listed potential threats to

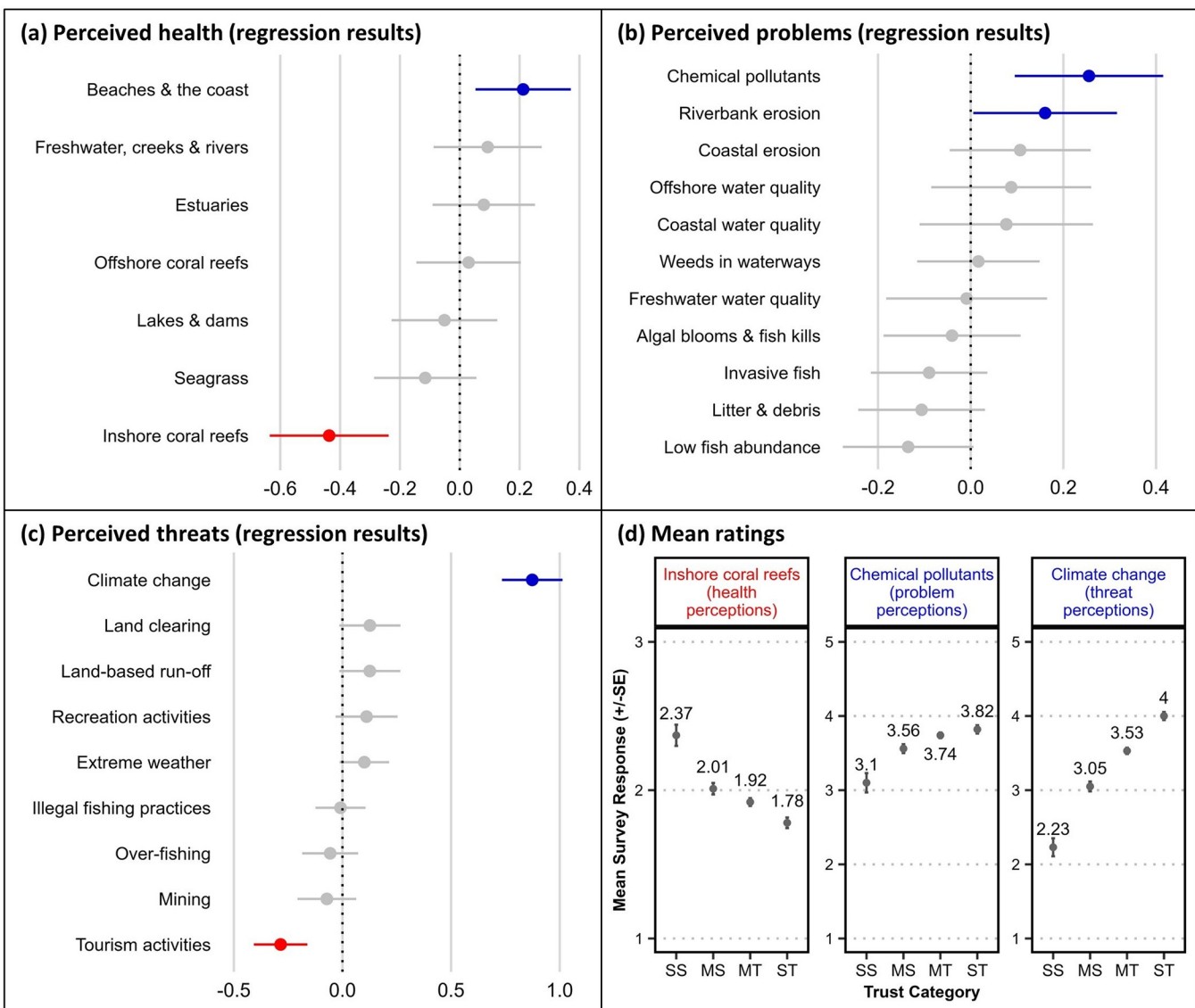

**Fig 4. Relationships between survey respondents' *trust in science* and their perceptions of waterway health, problems and threats.** Panels a–c show ordinal regression test results plotting survey respondents' *trust in science* with (a) perceptions of waterway health (n = 767), (b) perceptions of waterway problems (n = 1,251) and (c) perceptions of waterway threats (n = 1,877). Regression coefficients [dots] and standard error [SE] bars show statistical significance of the relationship where intersection of the SE bar with zero indicates lack of significance. Blue colouring indicates a significant positive relationship and red colouring indicates a significant negative relationship. Panel (d) shows the mean rating scores (±SE) from four groups with differing stated trust in science (SS = strongly sceptical, MS = mildly sceptical, MT = mildly trusting, ST = strongly trusting) for selected survey items with a significant regression result.

waterways there was no significant relationship between perceptions of their severity and *trust in science*. Despite a significant regression result for *tourism activities*, a comparison of the mean ratings showed no distinguishable difference between the four trust groups (S10 Table).

## Descriptive typology

Distinguishing characteristics of the four trust groups, drawn from our results above, are summarised below (Table 2). In this descriptive summary we have included only those predictors that were found to be statistically significant *and* that could be described as qualitatively distinct. I.e., some statistically significant predictors are excluded due to insufficiently distinct

**Table 2. Summary typology with distinguishing characteristics of four groups with differing stated levels of trust in the science underpinning waterway health and management in the GBR region (n = 1,877).**

**Strongly Sceptical (SS); n = 141 respondents, representing 8% of survey sample**

• Mostly male (69%), with longer lived experience in the region than other groups (mean = 31 years).
• Higher proportion (16%) employed in agricultural sector than other groups. Higher proportion (11%) employed in mining sector than MT and ST groups.
• Less likely to value First Nations heritage in their region than other groups.
• Likely to hold a perception that waterway management decisions are unfair.
• Likely to feel they are unable to provide input into management of waterways.
• Likely to feel less personal responsibility for improving waterway health than MT and ST groups.
• Likely to be less motivated to contribute to improving waterway health than MT and ST groups.
• Likely to have a lower sense of personal efficacy (belief that their actions can make a difference) towards improving waterway health than MT and ST groups.
• Likely to perceive inshore coral reefs as being in better health than other groups.
• Likely to perceive chemical pollutants in waterways as somewhat less of a problem than other groups.
• Likely to perceive climate change as a minor threat or not a threat to their region's waterways.

**Mildly Sceptical (MS); n = 438 respondents, representing 23% of survey sample**

• Balanced gender representation, with longer lived experience in the region than MT and ST groups (mean = 26 years).
• Higher proportion (11%) employed in mining sector than MT and ST groups.
• More likely than not to hold a perception that waterway management decisions are unfair.
• Less likely to feel they can provide input into management of waterways than MT and ST groups.
• Likely to feel less personal responsibility for improving waterway health than MT and ST groups.
• Likely to be less motivated to contribute to improving waterway health than MT and ST groups.
• Likely to have a lower sense of personal efficacy (belief that their actions can make a difference) towards improving waterway health than MT and ST groups.
• Likely to perceive climate change as a minor to moderate threat to their region's waterways.

**Mildly Trusting (MT); n = 839 respondents, representing 45% of survey sample**

• Balanced gender representation.
• Lower proportion (11%) employed in primary industries than SS and MS groups.
• Lower proportion (22%) dependent on regional waterways for household income than SS and ST groups.
• Likely to perceive climate change as a moderate to serious threat to their region's waterways.

**Strongly Trusting (MT); n = 459 respondents, representing 24% of survey sample**

• Slightly higher proportion are female (59%).
• Lower proportion (11%) employed in primary industries than SS and MS groups.
• More likely to value First Nations heritage in their region than other groups.
• More likely to participate in wildlife watching and appreciating nature than other groups.
• More likely to contribute to environmental monitoring than other groups.
• Likely to feel more personal responsibility for improving waterway health than other groups.
• Likely to be more motivated to contribute to improving waterway health than other groups.
• Likely to have higher sense of personal efficacy (belief that their actions can make a difference) towards improving waterway health than MT and ST groups.
• Likely to perceive inshore coral reefs as being in poorer health than other groups.
• Likely to perceive climate change as a serious threat to their region's waterways.

differences in the mean ratings of the four trust groups. For example, while there was a significant regression result for ratings of the *threat posed by tourism activities* and *trust in science*, the mean ratings for the four trust groups differed by only 0.03 on the five-point scale, indicating little dissimilarity in this perception (S10 Table). Similarly, the mean ratings for waterways' *existence value* (all above 8.2 on the 10-point scale), indicate that all four groups value this very highly, despite the significant regression test result (Fig 1B).

## Discussion

### Prevalence and characteristics of science mistrust in the GBR region

Our study found that 31% of survey respondents, representing residents of four major catchments in the GBR region, were mistrusting of the science underpinning waterway health and

management to some degree. Considering the time since our survey data were collected (November 2021) and indications from a more recent study of the same population [107], it is possible that this proportion has grown, mirroring a trend in Australia, the USA, and numerous other countries [121]. In recent years, cross-national studies have sought to identify drivers of public mistrust in science, in the context of resurging populist politics, the COVID-19 pandemic, and growth in misinformation presented in news media and across social media [42,121,122]. While it was beyond the scope of our study to identify specific drivers of science mistrust in the GBR region, it is likely that multiple factors have contributed, and that science sceptical attitudes and beliefs are not homogeneous. Relevant to our case study population, and considering the relatively high proportion of SS respondents employed in the agriculture sector (Table 1), factors contributing to the erosion of trust in the GBR region could include resentment among landowners to regulatory land use changes (intended to improve GBR water quality and underpinned by the GBR Scientific Consensus Statement on land use impacts on GBR health [81]), as well as climate change misinformation and denial narratives in social media [98]. An empirical assessment of social media posts by Lubicz-Zaorski et al. [98] found that a small group of politically aligned actors were responsible for a campaign of misinformation specifically about the GBR, with an apparent motive to erode public support for Australian climate change policy. While further research into the drivers of science mistrust is needed, knowing the factors associated with this mistrust can be useful for understanding how it is manifested in the regional context, and for understanding viewpoints that are shared or divergent when seeking constructive engagement.

## Characteristics of mistrust—disempowerment

Some of the distinguishing characteristics of the SS and MS groups (in Table 2) warrant further consideration by both scientists and natural resource managers. Respondents in these groups were likely to perceive that waterway management decisions are unfair, that they are unable to provide input into management, and perhaps consequently–they felt less responsible, less motivated and had a lower sense of personal efficacy in contributing to the improvement of waterway health. A sense of moral responsibility and of personal efficacy have been established as key motivating factors in people's adoption of stewardship behaviours [123]. However, the extent to which the empowerment of individuals mediates their trust in science and institutions in governance systems is a topic of ongoing research and debate. While some studies have found that higher levels of citizen participation and information transparency can yield greater trust among citizens [80], others have argued that trust-eroding suspicion and blame can arise as a perverse consequence of increased information sharing [124], and that contextual factors such as citizens' knowledge of relevant issues and predisposition to trust play an important mediating role [125]. Nonetheless, the apparent disempowerment of science-sceptical residents in the GBR region may be worthy of efforts to better understand and address real or perceived barriers to their participation in waterway governance and stewardship.

## Characteristics of mistrust—climate change scepticism

While perceptions of ecosystem health, problems and threats were mostly unrelated to respondents' trust in science (Fig 4), the divergent perceptions of climate change as a threat are characteristic of a broader societal phenomenon. While individual belief systems and forms of climate change scepticism and denialism vary, an underlying characteristic is the rejection of scientific findings that conflict with the individual's interests, beliefs and/or worldview [126–128]. At the extremes, such views can become entrenched in an individual's identity, political allegiance, and/or religion [128–130]. However, cognitive biases and the tendency to reject or

superficially manipulate conflicting evidence in favour of a preferred interpretation are universal flaws that can be expressed by individuals of any persuasion [128,131]. Proponents of climate change misinformation often exploit these psychological biases by appealing to people's identity, values, and belief systems, and often promote pseudoscience to create fake controversies and/or frame conspiracies (e.g. 'scientists falsifying results to obtain more funding') to elicit emotional reactions and to sow doubt and mistrust [127,131,132].

In the GBR context and region, climate change misinformation is pervasive and has been shown to be deployed opportunistically in response to emergent scientific reports and news about the Reef and its health [98]. A predominant misinformation narrative is that the Reef is 'in great health' and is unaffected by climatic and other anthropogenic pressures, such as poor water quality (ibid.). Conversely, media narratives arising from major disturbance events, such as the mass coral bleaching in 2016 and 2017, have tended to portray the Reef as 'dead' or 'dying' in a sensationalist manner [96,97,104]. Public risk perceptions can be influenced by 'risk events' and accompanying media representations, and in 2017 an increase was observed in the proportion of GBR residents and stakeholders who identified climate change as 'an immediate threat requiring action' [101,102]. While a potential counteractive effect of climate change misinformation in the GBR region has not yet been determined, repeat studies in 2021 and 2023 have shown that the proportion of residents who perceive climate change as an 'immediate threat' has decreased considerably [106,107]. Of additional concern is the potential effect of the opposing narratives 'in great health,' and 'dead' or 'dying', on individuals' personal efficacy and motivation to adopt pro-environmental behaviours. Both narratives serve to rationalise inaction, whether it be efforts to reduce carbon emissions or to improve coastal water quality, because acting is either unnecessary, or it's too late.

## Engaging with science mistrust

At a global scale, countering climate change and anti-science misinformation in the public sphere is seen increasingly as an imperative for deliberative democracies that seek to mitigate a worsening climatic outlook [133–135]. While communication techniques that debunk misinformation and 'inoculate' the public can reduce the influence of misinformation [136,137], public communication on its own is not sufficient to rebuild or maintain trust in science among citizens and stakeholders [12,14,18]. A healthy relationship between science, policy, and publics with optimal levels of trust requires integrity and credibility of both scientific and government institutions [12,15,16], and requires communities and stakeholders to be engaged in equitable processes and partnerships that recognise shared and diverse values, address shared goals, and build shared understandings of risks, opportunities, costs, and benefits [14,17,18,25].

The group characteristics summarised in our typology (Table 2) are useful to consider when engaging with communities and stakeholders in environmental management initiatives in the GBR region, and some of these characteristics may be relatable to other regions and contexts. But despite the differences between the four trust groups, more importantly, our results also reveal many similarities between science sceptical and science trusting residents. Encompassing their environmental values, uses and benefits, motivations, attitudes and perceptions, these similarities can be considered 'common ground' and can provide a useful basis for consensus-building discussions and message framing. Noting that individual viewpoints can and do vary, some key 'common ground' factors in our results (i.e. factors not associated with trust or mistrust in science) included:

i. *Shared appreciation of ecosystem values*, including biodiversity, existence, and icon values of regional waterways, such as the GBR (Fig 1A).

ii. *Shared recognition of the importance of ecosystem services*. For example, waterways supplying fish and seafood, supporting regional industry and economies, enabling social opportunities, recreation, lifestyle, aesthetic appreciation and contributing to personal wellbeing (Fig 2C and 2D).

iii. *Shared perceptions of community norms around environmental stewardship*, including levels of support and participation in some stewardship actions within the local community (Fig 3A and 3C).

iv. *Shared perceptions of the relative health of most aquatic ecosystems*, including freshwater habitats, estuaries and offshore waters (Fig 4A).

v. *Shared perceptions of many problems present in their region's waterways*, including erosion, the presence of weeds and invasive species, litter and debris, and reduced fish stocks (Fig 4B).

vi. *Shared perceptions of many threats to their region's waterways*, including over-fishing and illegal fishing, extreme weather, land clearing and land-based runoff (Fig 4C).

Insights about shared values and viewpoints, like those above, are often applied in communications and engagement efforts to demonstrate an understanding of others' perspectives, and common goals [66]. Numerous authors have studied and offer advice on communication techniques for public relations and organisational leadership that can help to establish a shared understanding of situation and risks, and to engender a commitment to shared goals and pathways to achieving them [e.g., 64,67]. Communications from relatable leaders and 'trusted messengers' (i.e. recognisable spokespeople with credibility among the target audience) can be influential in framing issues and building consensus [65,66,68]. However, as media and social media platforms have become increasingly fragmented, and as science-sceptical audiences have turned away from sources that report factual and scientific information, connecting such audiences with scientific or technical content has become increasingly difficult [31,32]. Communicating with science-sceptical audiences thus requires the use of a wider range of platforms, and interpersonal forms of communication stand a greater chance of resonating with audiences on polarised and technical subjects [32,65]. Fundamentally, trust arises from social relationships, when individuals and institutions demonstrate their competence, reliability, and trustworthiness [12,138].

Some anti-science views can be entrenched and are unlikely to change, especially when they are reinforced frequently by misinformation and peer networks [128,139]. Directly challenging or dismissing extreme viewpoints as flawed or invalid often results in alienation and estrangement [130,139]. Pursuing opportunities for respectful discourse on less contentious topics and shared viewpoints instead may yield more constructive and sustained engagement. For example, landowners who are sceptical of climate change and the effects of terrestrial runoff on inshore reefs may be more receptive to waterway management initiatives that seek to mitigate topsoil erosion and other impacts from extreme weather events such as floods or drought.

## Limitations, knowledge gaps and further research

The scope of our study was limited by using quantitative data derived from a survey that was designed for long-term monitoring of a broad selection of indicators relevant to 'human dimensions' of GBR regional waterways (i.e. social, economic, cultural and governance aspects). The primary metric for 'trust in science' was framed as a general concept, which did not allow for deeper investigation of the different types and component attributes of such trust (e.g., perceptions of the competence of scientists, of their motivations, and of systems

underpinning scientific integrity; as described by Stern & Baird [25]). Future research that explores such attributes may help to identify specific drivers of mistrust and their proportionate influence. Our study's description of trust characteristics at the macro scale can guide indepth, qualitative research that is necessary for understanding individual and smaller group characteristics [140,141].

Among the underlying drivers of mistrust in the science underpinning waterway health and management in our case study region, the extent to which such views are influenced by misinformation, and/or personal experience is of particular interest. Considering the misinformation campaign that specifically targets Reef and water quality science [98] and an observed trend of declining trust in the region's science and management institutions [107], further research that deconstructs this misinformation could be explored in controlled laboratory experiments, manipulating trust factors that are known to be important. Such research should be useful in counter-communications to inoculate the public against its corrosive effects (e.g. as reported by Cook and others [136,137]).

Long-term monitoring of community values and perceptions of environmental health, problems and threats remains important. As public risk perceptions can change in response to disturbance events and associated media representations [101,102], they may be similarly responsive to misinformation. Lastly, further research is needed into the effectiveness of communications that seek to rebuild public trust in science. While appeals to common values and other communication techniques have been studied extensively in the context of public relations and organisational leadership, there is still a paucity of empirical research on their influence on public trust in science.

## Conclusion

Our empirical case study in the GBR region achieves two things: (1) it contributes to an improved understanding of how science mistrust manifests in relation to aquatic ecosystem management at a regional scale, and (2) it provides insights on characteristics of science sceptical groups and on 'common ground' that can be applied by scientists, communicators, and resource managers when engaging with stakeholders and communities to build consensus on mutual goals and pathways, trust, and support to protect and restore habitats and vital ecosystem functions.

The management of waterways encompassing the Great Barrier Reef World Heritage Area and its adjacent catchments is at a pivotal stage, with significant government investments and scientific research focussed on reducing human pressures and enhancing ecosystem resilience to withstand impacts of increasing severity as the oceans and climate become warmer. The modern trend of increasing climate change denialism, anti-science sentiment, and misinformation represents a significant challenge to environmental policy in the GBR region and worldwide. Leaders, resource managers and scientists must contend with this challenge, and they face an increased impetus to uphold the integrity of science and its relationship with government policy, to counter misinformation, and to engage and empower communities in natural resource governance processes.

Our findings may challenge some assumptions and stereotypes about science scepticism but may confirm others. It should be somewhat reassuring to scientists and resource managers in the region that most residents are trusting of the science about waterway health and management. However, such levels of public trust are not guaranteed to be self-sustaining, and other results indicate widespread perceptions that aspects of community involvement in waterway management can be improved (S3 Table). It may also be reassuring then that there are many similarities (and abundant 'common ground') in environmental values and

perceptions shared among people who trust the science and those who do not. While a lack of trust in science may affect how one perceives threats to the environment, it does not necessarily affect how one sees the environment itself.

## Supporting information

**S1 Table. Results of ordinal regression models testing the relationship between survey respondents' 'trust [in] the science about waterway health and management' and predictor variables from five survey questions about respondent demography.**
(DOCX)

**S2 Table. Results of ordinal regression models testing the relationship between survey respondents' 'trust [in] the science about waterway health and management' and predictor variables from survey questions about values attributed to regional waterways, and mean rating scores (±SE) from four groups with differing stated trust in science (strongly sceptical, mildly sceptical, mildly trusting, strongly trusting) for each predictor variable.**
(DOCX)

**S3 Table. Results of ordinal regression models testing the relationship between survey respondents 'trust [in] the science about waterway health and management' and predictor variables from survey questions about perceptions of waterway governance, and mean rating scores (±SE) from four groups with differing stated trust in science (strongly sceptical, mildly sceptical, mildly trusting, strongly trusting) for each predictor variable.**
(DOCX)

**S4 Table. Results of ordinal regression models testing the relationship between survey respondents 'trust [in] the science about waterway health and management' and predictor variables from survey questions about recreational uses of regional waterways, and mean rating scores (±SE) from four groups with differing stated trust in science (strongly sceptical, mildly sceptical, mildly trusting, strongly trusting) for each predictor variable.**
(DOCX)

**S5 Table. Results of ordinal regression models testing the relationship between survey respondents 'trust [in] the science about waterway health and management' and predictor variables from survey questions about personal benefits derived from regional waterways, and mean rating scores (±SE) from four groups with differing stated trust in science (strongly sceptical, mildly sceptical, mildly trusting, strongly trusting) for each predictor variable.**
(DOCX)

**S6 Table. Results of ordinal regression models testing the relationship between survey respondents 'trust [in] the science about waterway health and management' and predictor variables from survey questions about respondents' participation in waterway stewardship actions, and mean rating scores (±SE) from four groups with differing stated trust in science (strongly sceptical, mildly sceptical, mildly trusting, strongly trusting) for each predictor variable.**
(DOCX)

**S7 Table. Results of ordinal regression models testing the relationship between survey respondents 'trust [in] the science about waterway health and management' and predictor variables from survey questions about respondents' motivation and capacity to participate in stewardship actions to improve the health of their region's waterways (i.e. stewardship**

enablers), and mean rating scores (±SE) from four groups with differing stated trust in science (strongly sceptical, mildly sceptical, mildly trusting, strongly trusting) for each predictor variable.
(DOCX)

**S8 Table. Results of ordinal regression models testing the relationship between survey respondents 'trust [in] the science about waterway health and management' and predictor variables from survey questions about perceptions of waterway health, and mean rating scores (±SE) from four groups with differing stated trust in science (strongly sceptical, mildly sceptical, mildly trusting, strongly trusting) for each predictor variable.**
(DOCX)

**S9 Table. Results of ordinal regression models testing the relationship between survey respondents 'trust [in] the science about waterway health and management' and predictor variables from survey questions about perceptions of waterway problems, and mean rating scores (±SE) from four groups with differing stated trust in science (strongly sceptical, mildly sceptical, mildly trusting, strongly trusting) for each predictor variable.**
(DOCX)

**S10 Table. Results of ordinal regression models testing the relationship between survey respondents 'trust [in] the science about waterway health and management' and predictor variables from survey questions about perceptions of threats to regional waterways, and mean rating scores (±SE) from four groups with differing stated trust in science (strongly sceptical, mildly sceptical, mildly trusting, strongly trusting) for each predictor variable.**
(DOCX)

## Acknowledgments

This research was conducted using data from the Social and Economic Long-Term Monitoring Program for the Great Barrier Reef (SELTMP; https://research.csiro.au/seltmp), collected in partnership with the Queensland Office of the Great Barrier Reef and World Heritage, and the five Regional Report Card Partnerships in the Great Barrier Reef catchment region: Wet Tropics Waterways, Dry Tropics Partnership for Healthy Waters, Mackay-Whitsunday-Isaac Healthy Rivers to Reef Partnership, Fitzroy Partnership for River Health, and Gladstone Healthy Harbour Partnership. The scientific results and conclusions, as well as any views or opinions expressed herein, are those of the authors and do not necessarily reflect the views of the Australian Government or the Queensland Government and their respective Ministers for the Environment.

## Author Contributions

**Conceptualization:** Matthew I. Curnock, Danielle Nembhard, Rachael Smith, Katie Sambrook, Elizabeth V. Hobman, Aditi Mankad, Petina L. Pert.

**Data curation:** Matthew I. Curnock, Petina L. Pert, Emilee Chamberland.

**Formal analysis:** Matthew I. Curnock, Danielle Nembhard.

**Funding acquisition:** Matthew I. Curnock, Aditi Mankad.

**Investigation:** Matthew I. Curnock, Petina L. Pert.

**Methodology:** Matthew I. Curnock, Elizabeth V. Hobman, Aditi Mankad, Petina L. Pert.

**Project administration:** Matthew I. Curnock, Aditi Mankad.

**Supervision:** Matthew I. Curnock.

**Validation:** Matthew I. Curnock, Danielle Nembhard.

**Visualization:** Matthew I. Curnock, Danielle Nembhard, Emilee Chamberland.

**Writing – original draft:** Matthew I. Curnock, Danielle Nembhard, Rachael Smith, Katie Sambrook, Elizabeth V. Hobman, Aditi Mankad.

**Writing – review & editing:** Matthew I. Curnock, Danielle Nembhard, Rachael Smith, Katie Sambrook, Elizabeth V. Hobman, Aditi Mankad, Petina L. Pert, Emilee Chamberland.

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
