## [Decision Letter · Decision Letter 0]

8 Jul 2024

PONE-D-24-08298Finding Common Ground: understanding and engaging with science mistrust in the Great Barrier Reef regionPLOS ONE

Dear Dr. Curnock,

Thank you for submitting your manuscript to PLOS ONE. After careful consideration, we invite you to submit a revised version of the manuscript that addresses the points raised during the review process.

We look forward to receiving your revised manuscript.

Kind regards,

Umberto Baresi, Ph.D.

Academic Editor

PLOS ONE

Journal Requirements:

"Funding was provided by the partnership between the Australian Government’s Reef Trust and the Great Barrier Reef Foundation, delivered in partnership with CSIRO, the Great Barrier Reef Marine Park Authority, and the Queensland Government’s Reef Water Quality Program. Collection of the survey dataset involved co-investment and in-kind support from the Queensland Office of the Great Barrier Reef and World Heritage, and the five Regional Report Card Partnerships in the Great Barrier Reef catchment region: Wet Tropics Waterways, Dry Tropics Partnership for Healthy Waters, Mackay-Whitsunday-Isaac Healthy Rivers to Reef Partnership, Fitzroy Partnership for River Health, and Gladstone Healthy Harbour Partnership."

"This research was conducted using data from the Social and Economic Long-Term Monitoring Program for the Great Barrier Reef (SELTMP; https://research.csiro.au/seltmp), with funding provided by the partnership between the Australian Government’s Reef Trust and the Great Barrier Reef Foundation, delivered in partnership with CSIRO, the Great Barrier Reef Marine Park Authority, and the Queensland Government’s Reef Water Quality Program. Collection of the survey dataset involved co-investment and in-kind support from the Queensland Office of the Great Barrier Reef and World Heritage, and the five Regional Report Card Partnerships in the Great Barrier Reef catchment region: Wet Tropics Waterways, Dry Tropics Partnership for Healthy Waters, Mackay933 Whitsunday-Isaac Healthy Rivers to Reef Partnership, Fitzroy Partnership for River Health, and  Gladstone Healthy Harbour Partnership. The scientific results and conclusions, as well as any views or opinions expressed herein, are those of the authors and do not necessarily reflect the views of the Australian Government or the Queensland Government and their respective Ministers for the Environment. The authors declare no conflict of interest."

"Funding was provided by the partnership between the Australian Government’s Reef Trust and the Great Barrier Reef Foundation, delivered in partnership with CSIRO, the Great Barrier Reef Marine Park Authority, and the Queensland Government’s Reef Water Quality Program. Collection of the survey dataset involved co-investment and in-kind support from the Queensland Office of the Great Barrier Reef and World Heritage, and the five Regional Report Card Partnerships in the Great Barrier Reef catchment region: Wet Tropics Waterways, Dry Tropics Partnership for Healthy Waters, Mackay-Whitsunday-Isaac Healthy Rivers to Reef Partnership, Fitzroy Partnership for River Health, and Gladstone Healthy Harbour Partnership."

**Additional Editor Comments:**

This manuscript has received glowing reviews, and it will provide a valuable contribution to the literature and to our journal's readership.

Please consider the feedback provided by Reviewer #2. I would suggest the authors to consider providing a bit of information on the Australian political context in relation to the Australian Great Barrier Reef.

Thank you for choosing PLOS One for publishing this high-quality piece of research.

Reviewers' comments:

Reviewer's Responses to Questions

**Comments to the Author**

1. Is the manuscript technically sound, and do the data support the conclusions?

Reviewer #1: Yes

Reviewer #2: Yes

2. Has the statistical analysis been performed appropriately and rigorously? 

Reviewer #1: Yes

Reviewer #2: Yes

3. Have the authors made all data underlying the findings in their manuscript fully available?

Reviewer #1: Yes

Reviewer #2: Yes

4. Is the manuscript presented in an intelligible fashion and written in standard English?

Reviewer #1: Yes

Reviewer #2: Yes

5. Review Comments to the Author

Reviewer #1: This manuscript is both comprehensive and thorough its content. Having read through and continually had comments I had written addressed by the next page or section read, I have no comments to submit beyond stating that I think this paper should be accepted. The work is insightful, meets the scope of PLOS ONE and I think would be valued by the journals readership.

Reviewer #2: This manuscript describes the results of a large public survey in Australia in the region near the Great Barrier Reef. In particular findings explore public trust in scientific natural resource management as it relates to other attitudes, values, and individual differences. Overall, I found the paper interesting and well written. The methods were appropriate, sound, and well described.

My only note is that some further treatment of how the key variables of values, attitudes toward civic institutions, and trust in science may be related in this region to partisan political identity. The authors touch on these subjects in the intro and discussion. Particularly given the association with some demographic data (gender, industry of employment), the importance of one's identifying with the values and attitudes of one's social group feels like it deserves further consideration here. In the US at least, trust in the realm of policy has become increasingly partisan, making traditional communication strategies to overcome mistrust more complicated to say the least. Even just a bit more coverage of the political context in which management and communication is taking place may be helpful. You touch on some of this in lines ~288-297 for instance, but it left me wanting to know a bit more. For instance, what are the major sources of misinformation in this region specifically and what interests are they bringing to the table.

I'll emphasize, that I don't want to make it seem like I am asking the authors to write a different paper with different questions/aims/variables. I'd just like to to see a bit more concrete grounding for the context and discuss the role of that context in interpreting results and implications.

6. PLOS authors have the option to publish the peer review history of their article (what does this mean?). If published, this will include your full peer review and any attached files.

Reviewer #1: No

Reviewer #2: No

---

## [Author Response · Author response to Decision Letter 0]

17 Jul 2024

We thank the editors and reviewers for their careful review and feedback. We have responded to this feedback in the Response to Reviewers document, with corresponding changes made in the updated manuscript files. We are hopeful that the manuscript is now acceptable for publication. Kind regards, Matt (corresponding author).

**UPDATE 18th July 2024: EDITORS - Please see the 'Response to Reviewers' document for the updated Financial Disclosure statement, which contains a statement on the role of funders. The PLOS One web portal does not allow me to edit the original statement that appears in the auto-generated PDF. Kind regards, Matt.**

---

## [Editor Report · Decision Letter 1]

22 Jul 2024

Finding common ground: understanding and engaging with science mistrust in the Great Barrier Reef region

PONE-D-24-08298R1

Dear Dr. Curnock,

We’re pleased to inform you that your manuscript has been judged scientifically suitable for publication and will be formally accepted for publication once it meets all outstanding technical requirements.

Kind regards,

Umberto Baresi, Ph.D.

Academic Editor

PLOS ONE

Additional Editor Comments (optional):

On behalf of PLOS One, I would like to thank the authors for their efforts. The paper in its current shape is accepted for publication.
---

## [Editor Report · Acceptance letter]

8 Aug 2024

PONE-D-24-08298R1 

PLOS ONE

Dear Dr. Curnock, 

I'm pleased to inform you that your manuscript has been deemed suitable for publication in PLOS ONE. Congratulations! Your manuscript is now being handed over to our production team.

Kind regards, 

on behalf of

Dr. Umberto Baresi 

Academic Editor

PLOS ONE